# The Role of Sclerostin in Bone Diseases

**DOI:** 10.3390/jcm11030806

**Published:** 2022-02-02

**Authors:** Elias S. Vasiliadis, Dimitrios-Stergios Evangelopoulos, Angelos Kaspiris, Ioannis S. Benetos, Christos Vlachos, Spyros G. Pneumaticos

**Affiliations:** 13rd Department of Orthopaedics, School of Medicine, National and Kapodistrian University of Athens, KAT Hospital, 16541 Athens, Greece; ds.evangelopoulos@gmail.com (D.-S.E.); ioannisbenetos@yahoo.gr (I.S.B.); christosorto@gmail.com (C.V.); spirospneumaticos@gmail.com (S.G.P.); 2Laboratory of Molecular Pharmacology, Division for Orthopaedic Research, School of Health Sciences, University of Patras, 26504 Rion, Greece; angkaspiris@hotmail.com

**Keywords:** sclerostin, *SOST* gene, Wnt-signaling pathway, antisclerostin antibody, bone formation, osteogenesis

## Abstract

Sclerostin has been identified as an important regulator of bone homeostasis through inhibition of the canonical Wnt-signaling pathway, and it is involved in the pathogenesis of many different skeletal diseases. Many studies have been published in the last few years regarding sclerostin’s origin, regulation, and mechanism of action. The ongoing research emphasizes the potential therapeutic implications of sclerostin in many pathological conditions with or without skeletal involvement. Antisclerostin antibodies have recently been approved for the treatment of osteoporosis, and several animal studies and clinical trials are currently under way to evaluate the effectiveness of antisclerostin antibodies in the treatment of other than osteoporosis skeletal disorders and cancer with promising results. Understanding the exact role of sclerostin may lead to new therapeutic approaches for the treatment of skeletal disorders.

## 1. Introduction

In recent years, the research on signaling pathways regulating bone formation has led to the identification of potential targets for the management of skeletal diseases. Sclerostin was first described in the pathogenesis of two rare disorders, sclerosteosis [1] and van Buchem’s disease [2]. Sclerostin inhibits the canonical Wnt-signaling pathway, and through this action, it controls bone formation by osteoblasts [3]. Numerous research papers underline sclerostin’s involvement in the pathogenesis of many skeletal disorders. Antisclerostin antibodies have recently been approved for the treatment of osteoporosis [4,5], and several clinical studies are currently under way to evaluate the effectiveness of antisclerostin antibodies in the treatment of other than osteoporosis skeletal diseases and cancer. The present review summarizes the existing knowledge of sclerostin’s role in the pathogenesis of numerous skeletal diseases and its role as a potential target for treatment.

## 2. Regulation of Sclerostin

Although sclerostin is mainly secreted by osteocytes [6], it has also been found in chondrocytes [6] and in osteoclasts [7]. With the exception of bone, sclerostin has been found in the lungs, kidneys, and liver [8] as well as in the epididymis, pyloric sphincter, cerebellum, and the embryonic hand [9]. Elevated serum sclerostin levels were found in vascular calcifications in humans with and without renal disease [10].

Sclerostin is encoded by the *SOST* gene and is considered to be a Bone Morphogenetic Protein (BMP) antagonist, possibly by inhibiting BMP’s binding to its receptors [6]. By binding to the Wnt LRP 5/6 coreceptors, sclerostin inhibits the canonical Wnt-signaling pathway (Figure 1) and advocates differentiation of mesenchymal stem cells to osteoblasts.

Inactivation of the canonical Wnt-signaling pathway in mature osteoblasts/osteocytes reduces osteoprotegerin (OPG), the decoy receptor for receptor activator of nuclear factor kappa-B ligand (RANKL) and enhances osteoclast differentiation and bone resorption [11]. When the sclerostin antibody was administered to ovariectomized rats, bone-formation markers were increased, and bone-resorption markers were decreased, suggesting that sclerostin inhibition of the canonical Wnt-signaling pathway favors bone gain by the simultaneous enhanced bone formation and reduction in bone resorption [12]. In a knock-in mouse model expressing a dominant active (da) β-catenin in osteocytes (daβcat^Ot^ mice), both OPG and RANKL are increased, resulting in increased bone resorption and an elevated osteoclast number [13]. When analyzing RANKL expression in the osteocytes of mice overexpressing *SOST* (dentin matrix acidic phosphoprotein 1 (*DMP1*) *SOST* transgenic mice), RANKL was increased and OPG was decreased. In *SOST* knock-out mice, there is an increase only in OPG and not in RANKL, suggesting that *SOST* expression is required to increase RANKL by the canonical Wnt-signaling pathway. Neutralization of the sclerostin activity with the antisclerostin antibody in daβcat^Ot^ mice resulted in the elevation of OPG expression and the reversion of an increased expression of RANKL to control levels, showing that increased RANKL induced by osteocytic activation of β-catenin requires sclerostin function. These findings imply that activation of the canonical Wnt-signaling pathway has different effects in osteocytes against osteoblasts regarding bone formation, as OPG is increased in both cell lines, whereas RANKL is increased only in osteocytes [13], and it could be a possible explanation for the uncoupling of bone formation and resorption after antisclerostin-antibody treatment, although the direct effect of antisclerostin antibodies on the osteoclast lineage cells is not known. Both Frizzled-8 (FZD8) and osteoclast precursor specific β-catenin knock-out mice showed osteopenia due to enhanced bone resorption, implying that osteoclast suppression through activation of the canonical Wnt-signaling pathway exists in an OPG-independent manner and probably acts on osteoclast precursors [14].

In the *SOST* gene, there are two sites where different transcription factors are binding and promote sclerostin expression. Evolutionary conserve region 5 (ECR5) is the site where myocyte enhancer factor (Mef2c) is binding and triggers sclerostin expression. Failure of this mechanism is the cause of Van Buchem’s disease or hyperostosis corticalis generalisata, where the absence of sclerostin results in uncontrolled bone formation. Hyperostosis, which is found mainly in the skull, mandible, and the ribs, cause deafness and paralysis of the face due to nerve entrapment within the excessive bone. [2,15]. The second site in the *SOST* gene is the upstream promoter region, where runt-related transcription factor 2 (Runx2) binds [16] and suppresses sclerostin expression [17]. Similarly, sirtuin 1, a histone deacetylase (HDAC), which is increased in hypoxia, suppresses the *SOST* gene in osteocytes by deacetylation of histone 3 at the *SOST* promoter [18,19] (Figure 2).

*SOST*-gene expression is increased by glucose and glycation end products [20] as well as in mice with diabetes type 1 [21], while androgens and estrogens suppress *SOST*-gene expression [22]. Glucocorticoids increase the expression of the *SOST* gene [23,24], and sclerostin deficiency has a protective effect against bone loss from glucocorticoids, mainly because of the anticatabolic effect of the canonical Wnt-signaling pathway and not because of its anabolic effect [25] (Figure 2). In *SOST* knock-out mice, the administration of glucocorticoids results in a decrease in OPG and an increase of the RANKL/OPG ratio. Treatment with the antisclerostin antibody inhibits bone resorption through OPG upregulation [22].

Evidence for the role of vitamin D in sclerostin regulation is contradictory. Vitamin D either provokes [26] or reduces sclerostin expression [27] (Figure 2).

SMADs, by binding to ECR5, mediate regulation of the *SOST*-gene expression through transforming growth factor-β (TGF-β), actinin-A, and BMPs [28]. The interleukin-6 (IL-6) cytokine family proteins, such as leukemia inhibitory factor (LIF), oncostatin (OSM), and cardiotropin-1 (CT-1), also suppress the expression of sclerostin [15] (Figure 2).

Dickkopf-1 (DKK1), another inhibitor of the canonical Wnt-signaling pathway, complements sclerostin to avoid excessive osteoblast differentiation. LRP5/6 has four BP (β-propeller) domains, which are regions where a Wnt-ligand is binding. DKK1 binds to the BP1 and BP3 domains, while sclerostin binds to the BP1 domain. Serum sclerostin concentration was found to be increased in DKK1 knockout mice, which implies that transcription of *SOST* is activated by the upregulation of the canonical Wnt-signaling pathway caused by the reduced DKK1 expression [13] (Figure 2). The activation of the canonical Wnt-signaling pathway results in increased bone formation and simultaneous upregulation of the expression of sclerostin and DKK1, suggesting that these agents act as a negative feedback mechanism to excessive bone gain [12].

Serum sclerostin levels were measured higher in older women, but *SOST* mRNA levels were found equal [29]. A possible explanation is that tissues other than bone may contribute to the elevation of sclerostin levels in mature age [29] (Figure 2).

The parathyroid-hormone (PTH) signals downregulate *SOST*-gene expression by inhibiting the activity of HDAC4/5 kinase (salt-inducible kinases—SIK), which results in the increase of dephosphorylated HDAC and its translocation into the nucleus. Nuclear dephosphorylated HDAC forms a complex with Mef2 and prevents binding of Mef2 with elements located in the ECR regulatory region [30]. In osteocyte-specific PTH/PTHrP receptor knock-out mice, *SOST*-gene expression is increased, and bone mass is decreased, suggesting that the anabolic effect of PTH is partly mediated by the suppression of sclerostin [31]. Intermittent administrations of the parathyroid hormone increase the osteoblast population by promoting proliferation, inhibiting apoptosis, and activating lining cells towards osteoblast differentiation [32]. Osteocytes have been recognized as target cells of the parathyroid hormone in bone [31,33], where it inhibits the *SOST* gene and consequently, the expression of sclerostin [34]. The parathyroid hormone, through binding to its receptors in osteocytes, downregulates sclerostin and exerts its anabolic effect by increasing bone formation [35] (Figure 2), and it regulates the expression of additional genes playing a role in bone homeostasis, such as RANKL, fibroblast growth factor 23 (FGF23), and OPG [30]. In mice with a deficient osteocytic parathyroid hormone receptor, mechanical loading did not decrease *SOST*-gene expression, implying that bone formation is stimulated by the parathyroid hormone through inhibition of sclerostin and upregulation of the canonical Wnt-signaling pathway [36]. *SOST* knock-out mice showed an increased anabolic effect of the parathyroid hormone in bone [37], while in mice overexpressing the human *SOST* gene, the parathyroid hormone did not increase bone formation [38]. In contrast, the parathyroid hormone showed significant anabolic effects in double transgenic *DMP1-SOST* mice, which also overexpress the human *SOST* gene. These findings imply that downregulation of *SOST*-gene expression is not necessary for the anabolic effect of the parathyroid hormone in bone, while it is essential for the effects of mechanical loading [30].

## 3. Role of Sclerostin in Skeletal Diseases

### 3.1. Osteoporosis

Osteoporosis is characterized by lower bone mineral density due to an imbalance between bone formation and bone resorption. Bone remodeling is disrupted at a cellular level, with bone resorption by osteoclasts exceeding bone formation by osteoblasts. Furthermore, in postmenopausal women, decreased estrogen levels stimulate osteoblast apoptosis and promote osteoclast activity [39]. Estrogens, together with the Wnt-signaling pathway upregulation, promote osteogenic differentiation [40]. The elevation of *SOST*-gene expression in osteocyte-specific estrogen receptor knock-out mice implicates the role of estrogens in regulation of sclerostin secretion by osteocytes [41]. Estrogen deficiency results in increased sclerostin expression. Serum sclerostin was found higher in postmenopausal than premenopausal women, and the administration of selective estrogen receptor modulators (SERMs) for the treatment of osteoporosis reduced serum sclerostin levels [42]. Moreover, sclerostin expression was found elevated in older women [8]. Sclerostin, as an inhibitor of the Wnt-signaling pathway, prevents osteoblastogenesis and OPG production, suppresses osteoclast-mediated bone formation, and increases bone resorption by stimulating RANKL expression from osteocytes [43]. Considering its role in bone homeostasis, targeting sclerostin with the antisclerostin antibody was evidenced as an effective treatment of osteoporosis [44]. In four phase III clinical trials, the administration of the antisclerostin monoclonal antibody, romosozumab, increased bone density, stimulated bone formation, and suppressed bone resorption [45]. The risk of vertebral and nonvertebral fractures was also found to be significantly lower [45,46]. A 12-month efficacy of romosozumab showed a 73% lower incidence of new vertebral fractures and a 36% lower incidence of clinical fractures (nonvertebral and symptomatic vertebral fractures) when compared with the placebo [45]. At 24 months in patients treated with romosozumab followed by alendronate, a 48% lower risk of new vertebral fractures, a 27% lower risk of clinical fractures, and a 38% lower risk of hip fractures were shown when compared with patients who received alendronate alone [46]. The anabolic effect of romosozumab, as it is demonstrated by the bone-turnover marker, procollagen type 1 intact N-terminal propeptide (P1NP), is significantly higher during the first year of treatment [45,47], followed by a slow decrease thereafter [45]. C-terminal telopeptide (CTX), a marker of bone resorption, showed a faster decrease and stabilized at around 50% of its highest measurement after the first year of romosozumab treatment [45]. Although sclerostin is mainly an osteocyte-derived protein [6], and antisclerostin antibodies primarily target bone, there is concern regarding romosozumab’s cardiovascular side effects [46,48], hypocalcemia [45], and oncogenesis [49]. In phase III clinical trials, romosozumab was associated with atrial fibrillation [46] and some serious cardiovascular adverse events, such as cardiac ischemic events, cerebrovascular events, heart failure, and peripheral vascular ischemic events [48]. Blosozumab, another humanized antisclerostin monoclonal antibody, showed a significant increase of bone mineral density in the spine and hip in postmenopausal women, although frequent injection-site reactions were reported. Biochemical markers of bone formation were elevated, and markers of bone resorption decreased [50].

Administration of the antisclerostin antibody in ovariectomized rats increased DKK1 expression [12], and osteocyte-specific DKK1 and *SOST* double knock-out mice demonstrated high bone mass [51]. Based on these findings, dual inhibition of sclerostin and DKK1 by administration of a bispecific antibody increased bone formation and improved fracture healing in rodents [52], suggesting that a simultaneous inhibition of sclerostin and DKK1 may result in superior results in osteoporosis treatment.

### 3.2. Glucocortosteroid Induced Bone Loss

Glucocorticoid-induced osteoporosis pathogenesis involves increased osteoclastogenesis due to an increased RANKL/OPG ratio, reduced osteoblast differentiation, increased apoptosis of osteoblasts, decreased matrix synthesis and osteocytes, and increased autophagy [53,54]. Glucocorticoids inhibit numerous genes of the canonical Wnt-signaling pathway through the increase of *SOST*-gene expression [22], although the exact mechanism is not clear. The lack of *SOST*-gene expression minimizes bone loss, which is aggravated by glucocorticoids by switching the canonical Wnt-pathway to an anticatabolic signaling [22]. Thus, targeting sclerostin with an antisclerostin antibody could be beneficial for the treatment of glucocorticoid-induced osteoporosis [54].

Glucocorticoid excess in *SOST* knock-out mice or in mice treated with the antisclerostin antibody inverted the rise of the RANKL/OPG ratio and demonstrated a decrease in bone resorption without altering bone formation [22]. The administration of antisclerostin antibodies in mice with glucocorticoid-induced osteoporosis increased bone mass [55], although after treatment with glucocorticoids they presented higher bone loss than controls, suggesting that there must be additional pathogenetic mechanisms in glucocorticoid-induced osteoporosis [56]. *SOST*-deficient mice treated with glycocorticoids alone showed higher bone-formation markers (osteocalcin and P1NP) and lower bone resorption markers (tartrate-resistant acid phosphatase (TRAP5b) and CTX-1) than mice treated with a combination of glucocorticoids and antisclerostin antibodies [25,56].

In humans, serum sclerostin levels were significantly reduced after glucocorticoid treatment [55]. In a patient with Van Buchem disease, the administration of glucocorticoids halted new bone formation, suggesting that glucocorticoids restrain bone formation due to sclerostin deficiency in humans [57].

### 3.3. Osteonecrosis

In patients with nontraumatic osteonecrosis of the femoral head, serum sclerostin levels were lower than controls, and the decrease was related to the stage of osteonecrosis, meaning the higher the stage, the lower the levels of serum sclerostin that were measured [58]. Additionally, sclerostin expression in the bone sections of femoral heads with osteonecrosis was significantly reduced compared to controls. Patients with severe radiological imaging of osteonecrosis or patients at the post-collapse stage appeared with statistically significant lower sclerostin expression. The above findings were irrelevant to the etiology of nontraumatic osteonecrosis of the femoral head [58]. Sclerostin’s downregulation was attributed to increased osteocyte apoptosis [59]. Low sclerostin expression causes enhanced osteoblast activity, which affects the balance between osteoblast and osteoclast in bone and accelerates the collapse of the femoral head [58].

Osteonecrosis of the jaw has been reported in two patients as an adverse effect of osteoporosis treatment with the antisclerostin antibody, romosozumab [45]. Blocking sclerostin has an antiresorptive function through osteoclast inhibition, and thus, it may cause osteonecrosis of the jaw. Administration of the antisclerostin antibody to rats with periodontitis did not reproduce the above side effect [60].

### 3.4. Osteoarthritis

*SOST*-gene expression has been detected in hypertrophic chondrocytes [61] and was found elevated in degenerative cartilage [62]. Interestingly, sclerotic subchondral bone in osteoarthritis exhibits decreased sclerostin expression [62,63], possibly because of increased mechanical forces. These findings suggest that sclerostin is involved in osteoarthritis pathogenesis. Chondrocytes isolated from osteoarthritic joints in animal models as well as in humans, demonstrated increased the canonical Wnt-signaling [62,64] and activation of β-catenin in normal chondrocytes resulted in an osteoarthritic phenotype in mice [65]. Given that upregulation of the canonical Wnt-signaling pathway in chondrocytes deteriorates osteoarthritis, sclerostin expression may have a protective role. Low sclerostin levels in aging mice, or the administration of the antisclerostin antibody in rats with surgically induced osteoarthritis did not cause any further damage in articular cartilage, despite the Wnt-signaling upregulation [65]. In contrast, *SOST* knock-out mice developed severe osteoarthritic changes after medial meniscus destabilization but with no osteophyte formation [66], suggesting a possible protective role of sclerostin in osteoarthritis. This conflicting response to the cartilage breakdown between *SOST* knock-out mice and the antisclerostin-antibody treatment is probably due to insufficient diffusion of sclerostin in articular cartilage and implies that an additional trigger is required to initiate osteoarthritis [64,66]. *SOST* knock-out mice as well as the antisclerostin antibody administration appear to upregulate DKK1, which may explain the inadequate inhibition of sclerostin on articular cartilage [52]. The effects of the antisclerostin antibody in joints raise some concerns for its use in osteoporosis treatment.

Chondrocytes harvested from osteoarthritic joints show an increased expression of sclerostin [62], in contrast to normal chondrocytes, which do not express sclerostin [3]. Furthermore, articular chondrocytes which were harvested during total knee replacement surgery appeared with an increased *SOST*-gene expression due to hypomethylation of the CpG region of the *SOST* promoter [67].

Patients with osteoarthritis have significantly lower levels of serum sclerostin than healthy controls [68], and a decrease of osteocyte sclerostin leads to increased bone formation, which explains the protective effect of osteoarthritis to hip fractures [69].

Targeting the Wnt-signaling pathway with molecules that inhibit its degenerative effect on articular cartilage in rats showed promising results [70,71]; therefore, clinical trials are needed for the development of novel therapies for osteoarthritis.

### 3.5. Rheumatoid Arthritis

Rheumatoid arthritis is characterized by decreased bone mass, bone and cartilage erosions, and systemic inflammation and is associated with increased serum sclerostin levels [72] and reduced Wnt signaling [73]. Several inflammatory cytokines from the synovial tissue in rheumatoid arthritis, such as tumor necrosis factor-α (TNF-α) and IL-6 trigger an increased expression of RANKL and matrix metaloproteinases (MMPs) [74]. TNF-α also increased DKK1 secretion from synoviocytes in rheumatoid arthritis, which upregulates MKK3-p38 and inhibits the canonical Wnt-signaling pathway [75]. Wnt5a, a ligand of the Wnt-signaling pathway, is also overexpressed in rheumatoid arthritis patients [76]. Wnt5a promotes the expression of inflammatory cytokines in synovial tissue [77], induces chemokine production [78], and regulates MMP production [79], and therefore, is implicated in bone and cartilage degradation and is a promising therapeutic target for rheumatoid arthritis.

In rheumatoid arthritis, sclerostin was isolated from the synovial tissue, and it is secreted by fibroblast-like synoviocytes [73] and from osteoblasts [80]. Sclerostin’s role in rheumatoid arthritis is unclear, as there are contradictory findings from animal studies in the literature. In mice with TNF-α induced rheumatoid arthritis (human TNF-α transgenic mice), the administration of antisclerostin antibodies improved osteochondral erosions even though they had no effect on systemic inflammation [81]. In mice with collagen-induced rheumatoid arthritis, antisclerostin antibodies similarly did not improve systemic inflammation but had no effect on bony and cartilage erosions either, although they prevented systemic bone loss [82]. On the contrary, in TNFα-induced rheumatoid arthritis mice that do not express sclerostin (*SOST* knock-out mice or after antisclerostin antibody administration), a rapid deterioration of rheumatoid arthritis occurred [73], suggesting that sclerostin may have a protective role. A possible explanation for the protective role of sclerostin in TNFα-induced rheumatoid arthritic mice is that it inhibits p38/ERK (extracellular signal-regulated kinase) activity and RANKL expression in synoviocytes [73].

These contradictory results question the effect of antisclerostin-antibody therapy in rheumatoid arthritis, and further studies are required before such a treatment is utilized.

### 3.6. Ankylosing Spondylitis

In contrast to rheumatoid arthritis, sclerostin expression in ankylosing spondylitis is markedly reduced. The percentage of positive-staining osteocytes for sclerostin was extremely low, and it was not associated with the rate of osteocyte apoptosis [83]. Similarly, serum sclerostin levels in patients with ankylosing spondylitis were significantly lower than controls. This reduction in sclerostin levels was more pronounced in patients with ankylosing spondylitis who already developed syndesmophytes than in patients who did not [83]. It is not clear whether sclerostin downregulation is a cause or an effect of new bone formation. In mice with ankylosing spondylitis, sclerostin autoantibodies were detected, suggesting their possible contribution to its pathogenesis [84]. Syndesmophytes and ossification in ankylosing spondylitis is rather a remodeling procedure and occur after the initial phase of inflammation, which causes bone erosions [85]. Patients with low sclerostin levels are more susceptible to developing ankylosis [86]. Therefore, sclerostin levels could be used as a biomarker for the prediction of syndesmophyte formation in ankylosing spondylitis.

### 3.7. Bone Tumors

#### 3.7.1. Primary Bone Tumors

In osteoma, osteoid osteoma, osteoblastoma, and osteosarcoma, osteocytes and chondrocytes from both articular cartilage and growth plate express sclerostin, while osteoblasts in osteoid osteomas and osteoblastomas, did not express sclerostin [87]. Chondroblasts in chondroblastoma and hypertrophic chondrocytes in osteochondroma, especially those embedded in mineralized chondroid matrices, also expressed sclerostin [88]. Positive stains of sclerostin were detected in immature bone in tumor-like lesions, such as fibrous dysplasia and osteofibrous dysplasia. Fibrosarcoma, Ewing sarcoma, and giant cell tumors did not express sclerostin [87].

Most high-grade osteosarcomas and parosteal osteosarcomas express sclerostin after the binding of the Runx2 to the *SOST* gene upstream promoter region, although sclerostin’s exact role is not known [88].

#### 3.7.2. Bone Metastasis

Sclerostin was increased in the breast cancer metastatic cell line, MDA-MB 231, and in the nonmetastatic cell line, MCF-7, due to the abnormal overexpression of Runx2, which binds to the proximal promoter of the *SOST* gene. Elevated sclerostin levels in metastatic breast cancer suppress bone formation through inhibition of the Wnt-signaling pathway, implicating its role in the pathogenesis of osteolytic lesions [89]. Inhibition of the Wnt-signaling pathway caused by overexpression of DKK1 in breast cancer cell lines resulted in a significantly higher impact of bone metastases, as osteogenesis is impaired and osteoclastogenesis is enhanced through OPG downregulation [90].

In patients with prostate cancer, sclerostin levels were lower than in patients with benign prostatic hyperplasia [91]. Low sclerostin and noggin levels in combination with high BMP6 expression had a predictive value for prostate cancer progression and metastases [91]. Sclerostin, by restricting the Wnt-signaling pathway, inhibited prostate cancer migration and sclerostin deficiency resulted in increased prostate-cancer spread [92]. PC3 prostate cancer cells which overexpressed the *SOST* gene showed less osteolysis when injected into the femur of non-obese diabetic (NOD) scid gamma (NSG) mice and significantly less metastases when administered intravenously [92]. Serum sclerostin levels were elevated in patients with metastatic prostate cancer disease [93] and in patients who received androgen deprivation therapy [94]. These findings imply that sclerostin, which is secreted locally by tumor cells, has a protective role in the progression of prostate cancer through the Wnt-signaling pathway inhibition, and elevated serum sclerostin levels reflect the increased activity of osteoblasts and osteocytes at the osteoblastic lesions induced by prostate cancer.

#### 3.7.3. Multiple Myeloma

In multiple myeloma (MM), bone marrow is occupied by plasma cells and patients experience localized osteolytic lesions due to uncoupled bone remodeling, with bone resorption prevailing bone formation [95]. Among other mechanisms, osteocytes seem to play an important role in bone destruction in MM [96]. In MM there is an increased apoptosis of osteocytes, possibly due to the secretion of TNFα from cancer cells, as the anti-TNF antibody reversed osteocyte death [97]. Furthermore, the activation of the Notch-signaling pathway in MM enhances osteocyte apoptosis and results in the upregulation of RANKL and *SOST*-gene expression. Increased RANKL activates osteoclasts, while elevated sclerostin inhibits the Wnt-signaling pathway, resulting in increased bone resorption and suppressed bone formation, respectively [97]. Serum and bone marrow sclerostin levels are increased in MM patients [98,99], and elevated serum sclerostin levels are correlated with poor survival [100]. Other potential sources of sclerostin in MM are CD138+ cells [101] and osteoblasts [99]. Genetic deletion of the *SOST* gene was found to decrease osteolytic lesions in the immunodeficient mouse model by increasing osteoblasts and decreasing osteoclasts in MM bone [102]. In the same way, pharmacological inhibition of sclerostin with the antisclerostin antibody in immunocompetent mice reduced osteolysis and resulted in the elevation of bone-formation markers without altering tumor growth [103]. The prevention of osteolysis and simultaneous tumor suppression was achieved by injection of MM plasma cells together with the antisclerostin antibody [103]. These findings indicate that a combination of the antisclerostin antibody with antitumor agents may have beneficial effects in MM treatment. Further studies are needed to determine whether antisclerostin antibodies affect tumor progression in patients with MM.

### 3.8. Osteogenesis Imperfecta

Osteogenesis imperfecta involves a group of genetic disorders which cause multiple fragility fractures. There are five groups according to the genetics of the disorder. Group A includes defects in the type I collagen structure due to a mutation in the COLIA1 and COLIA2 genes, which encode alpha1 and alpha2 chains and account for nearly 90% of the patients. Groups B, C, D, and E include mutations in genes which are responsible for post-translational modification of collagen, for collagen folding and cross-linking, for abnormalities in mineralization, and for osteoblast differentiation, respectively [104]. Additionally, mutations of the Wnt1 ligand led to a rare form of osteogenesis imperfecta. A mutant Wnt1 decreases the nuclear translocation of β-catenin and downregulates the Wnt-signaling pathway [105]. The anabolic effect of antisclerostin antibodies proved efficient in reducing the number of axial fractures [106] as well as the number of long bone fractures in mice with type III osteogenesis imperfecta [107]. Setrusumab (BPS-804), an antisclerostin antibody, has recently being investigated in humans with moderate osteogenesis imperfecta. In a phase IIa trial, lumbar spine BMD increased, bone formation biomarkers were elevated, and bone resorption biomarkers decreased [108]. A phase III trial, measuring the strength and quality of bone using a special type of CT scanner after administration of setrusumab in adults with type I, III, and IV osteogenesis imperfecta, is currently in progress (NCT03118570).

### 3.9. Hypophosphatasia

Hypophosphatasia is a genetic disorder caused by the reduced activity of alkaline phosphatase due to mutations in its gene [109]. Its skeletal disorders consist of incomplete mineralization, low bone turnover, and inferior bone quality, which is considered as a type of osteomalacia and clinically results in pseudofractures, stress fractures, and diffuse muscle pain [110]. In theory, bone anabolic agents could reverse the above clinical manifestations of hypophosphatasia. In a phase IIa clinical trial, the administration of the antisclerostin antibody, BPS804, elevated serum levels of bone-formation markers, increased bone-specific alkaline phosphatase enzymatic activity, decreased serum levels of the bone-resorption marker, CTX-1, and increased lumbar-spine bone mineral density [111].

## 4. Limitations

The present study addresses the involvement of sclerostin in the pathogenesis of bone diseases based on studies that either measure tissue and circulating sclerostin or report the effectiveness of antisclerostin-antibody treatment. Alterations in sclerostin levels or any potential positive effect of antisclerostin antibodies does not imply that sclerostin is involved in the etiology of these disorders. Further studies are required to better understand the exact role of sclerostin in bone diseases.

## 5. Conclusions

Although sclerostin is expressed from different cell types, its major source of secretion is the osteocyte. Sclerostin exhibits a key role in skeletal homeostasis by inhibiting bone formation and promoting bone resorption. Pharmacologic inhibition of sclerostin enhances bone formation and shows positive outcomes in osteoporosis treatment. Beyond osteoporosis, sclerostin is involved in the pathogenesis of numerous other skeletal disorders. Understanding the exact mechanism of this involvement may result in novel therapeutic approaches for the treatment of skeletal diseases.

## Figures and Tables

**Figure 1 jcm-11-00806-f001:**
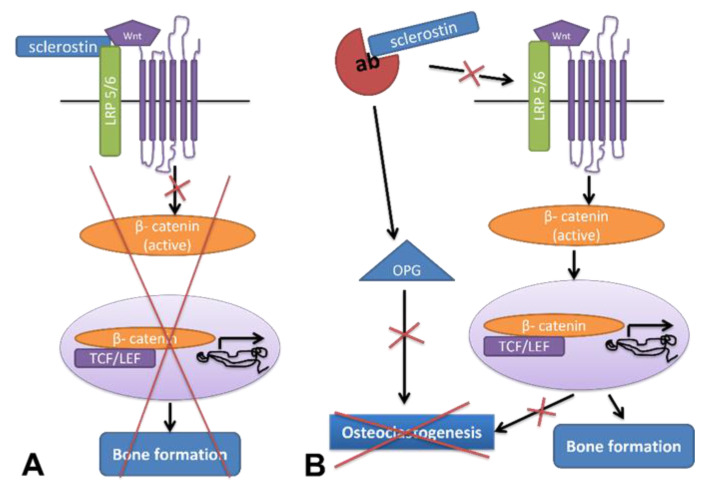
(**A**) Sclerostin inhibits canonical Wnt-signaling pathway through its binding to the Wnt LRP 5/6 coreceptors and leads to decreased bone formation. (**B**) When sclerostin antibody is administered, bone formation is increased through activation of Wnt signaling pathway and elevation of OPG expression, which inhibits osteoclastogenesis. LRP 5/6: low-density lipoprotein receptor-related protein 5/6, TCF: transcription factor, LEF: lymphoid enhancer-binding factor, OPG: osteoprotegerin, ab: antisclerostin antibody.

**Figure 2 jcm-11-00806-f002:**
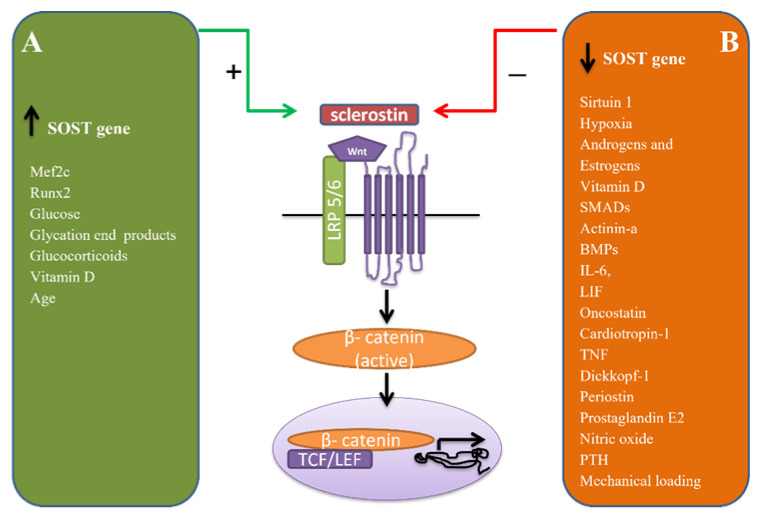
(**A**) Factors that upregulate *SOST*-gene expression. (**B**) Factors that down-regulate *SOST*-gene expression. Mef2c: myocyte-enhancer factor, Runx2: runt-related transcription factor 2, BMPs: bone morphogenetic proteins, IL-6: interleukin-6, LIF: leukemia inhibitory factor, TNF: tumor necrosis factor, LRP 5/6: low-density lipoprotein receptor-related protein 5/6, TCF: transcription factor, LEF: lymphoid enhancer-binding factor, PTH: parathyroid hormone.

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
