# Peer review of "The Role of Sclerostin in Bone Diseases"

_jcm, 2022, doi:10.3390/jcm11030806_

Round 1
Reviewer 1 Report
The manuscript is overall of good quality, well readable and actual.
2 comments:
- I missed better explanation for Van Buchen disease, please extend
- More exact data to decsribe effect of romosozumab on fracture risk are needed to be reported. Also, there is a paragraph describing cardiovasular harm in this medication-more explanation for this is needed.
Author Response
Point 1: I missed better explanation for Van Buchen disease, please extend.
Response 1: More information for Van Buchen disease are added in lines 81-84.
Point 2: More exact data to describe effect of romosozumab on fracture risk are needed to be reported.
Response 2: Additional details of romosozumab fracture efficacy are reported in lines 168-173.
Point 3: Also, there is a paragraph describing cardiovascular harm in this medication-more explanation for this is needed.
Response 3: More details regarding cardiovascular side effects of romosozumab from phase III clinical trials are provided in lines 180-183.
Reviewer 2 Report
Specific comments:
This is a well written and quite exhaustive review about sost and bone disorders. The review provides both mechanistic and clinical insights into sost biology. However it seems to imply that sost plays a pathophysiological role in all these disorders, whereas for most it remains uncertain. Even in the context of osteoporosis, there is no real direct evidence that sost plays a major role. Moreover, the authors make no attempt to clarify why anti-scl Ab have transient effects on bone formation but durable effects on bone resorption, as they imply both phenomena are driven by the Wnt-beta-catenin pathway, which is just impossible otherwise both the anabolic and anti-resorptive effects would be blunted simultaneously. Eventually, studies that have measured circulating sost in any disease dont provide any evidence for a role of sost in those diseases since there is great variability among the assays for circulating sost and no evidence that serum sost represents sost activity at the tissue level.
I suggest that the authors "nuance" their review to reflect these questions and limitations. Perhaps they could provide a table with the different diseases where they would grade the evidence for a direct pathophysiological role fo sost in human bone disorders.
Author Response
This is a well written and quite exhaustive review about sost and bone disorders. The review provides both mechanistic and clinical insights into sost biology.
Point 1: However, it seems to imply that sost plays a pathophysiological role in all these disorders, whereas for most it remains uncertain.
Response 1: The authors totally agree with this meaningful comment. A new section entitled “Limitations” is added at the end of the paper to emphasize the issues raised by the reviewer.
Point 2: Even in the context of osteoporosis, there is no real direct evidence that sost plays a major role. Moreover, the authors make no attempt to clarify why anti-scl Ab have transient effects on bone formation but durable effects on bone resorption, as they imply both phenomena are driven by the Wnt-beta-catenin pathway, which is just impossible otherwise both the anabolic and anti-resorptive effects would be blunted simultaneously.
Response 2: This is a very interesting comment. The authors tried to address the uncoupling of bone formation and resorption in osteoporosis treatment with romosozumab in the section “Regulation of Sclerostin”, lines 52-77. The authors do not imply that this uncoupling phenomenon is driven only by β-catenin Wnt pathway. One possible explanation could be a direct effect of romosozumab in the osteoclast lineage cells, which is currently not known. An attempt to clarify this is made in the added lines 71-73 and 175-178.
Point 3: Eventually, studies that have measured circulating sost in any disease don’t provide any evidence for a role of sost in those diseases since there is great variability among the assays for circulating sost and no evidence that serum sost represents sost activity at the tissue level.
Response 3: See response 1.
Point 4: I suggest that the authors "nuance" their review to reflect these questions and limitations. Perhaps they could provide a table with the different diseases where they would grade the evidence for a direct pathophysiological role of sost in human bone disorders.
Response 4: The authors added the new section with the title “Limitations” as stated above, instead of adding a table with grading of sclerostin’s role in the pathophysiology of bone diseases. Unfortunately, available data from the literature does not support any grading scheme.